Medical Imaging with Deep Learning 2024

# Myoblast Mutation Classification via Microgroove-Induced Nuclear Deformations

**Xingjian Zhang**[1,2]                           XINGJIAN.ZHANG@POLYTECHNIQUE.EDU

**Claire Leclech**[1]                             CLAIRE.LECLECH@POLYTECHNIQUE.EDU

**Bettina Roellinger**[1]                         BETTINA.ROELLINGER@POLYTECHNIQUE.EDU

**Catherine Coirault**[3]                         CATHERINE.COIRAULT@INSERM.FR

**Elsa D. Angelini**[2]                           ELSA.ANGELINI@TELECOM-PARIS.FR

**Abdul I. Barakat**[1]                           ABDUL.BARAKAT@POLYTECHNIQUE.EDU

[1] *LadHyX, École Polytechnique, Institut Polytechnique de Paris, France*

[2] *LTCI, Télécom Paris, Institut Polytechnique de Paris, France*

[3] *Sorbonne Université, INSERM UMRS-974 and Institut de Myologie, France*

## Abstract

Microgroove substrates induce 3D nuclear deformations in various adherent cell types. In this study, we explore the capacity of a CNN classifier to identify myoblast mutations through subtle differences in nuclear deformations on 2D fluorescence microscopy images. A large set of experimental images from immunostained nuclei screened on microgroove platforms is exploited. Leveraging ResNet-50 in a weakly-supervised setting, we present preliminary results to accurately classify healthy myoblasts from laminopathy-associated mutations. We achieved F1 scores of 0.99 and 0.94 at whole-image and patch levels evaluations. These results demonstrate the potential for microgroove screening as a functional diagnostic device of diseases characterized by aberrant nuclear deformations.

**Keywords:** microgrooves, cell mutation, nuclear deformation, image classification.

## 1. Introduction

Various diseases including laminopathies and certain types of cancer are associated with abnormal nuclear mechanical properties that influence cellular and nuclear deformations in complex environments (Zwerger et al., 2011). Recently, microgroove substrates designed to mimic the anisotropic topography of basement membranes have been shown to induce significant 3D nuclear deformations in various adherent cell types (Leclech et al., 2024). Importantly, these deformations appear to be different in muscle precursor (myoblast) cells derived from laminopathy patients from those derived from normal individuals. This underscores the potential of leveraging deep learning and computer vision to provide rapid and high throughput classification of cell mutations based on nuclear deformations.

In this study, we test the potential of a weakly-supervised CNN for classifying myoblast mutation in cells cultured on microgrooves. Our approach involves learning on image-level labels to capture the complex deformation patterns present across cell populations. We describe our image preprocessing and motivate our design choices to address the challenges associated with variability in nuclear deformation, cell densities, and small dataset size. We apply our method for the binary classification of healthy versus mutant myoblast cells.

## 2. Methods

**Data Collection.** Wild type (WT) control myoblasts from a healthy subject and myoblasts from a subject carrying the LMNA c.94_96delAAG, p.Lys32del mutation (hereafter denoted as $\Delta$K32) were derived from muscle biopsies. The cells were seeded at 100,000 cells/cm$^2$ on polydimethylsiloxane (PDMS) microgroove substrates prepared as detailed elsewhere (Leclech et al., 2022) (schematic in Fig. 1A). The microgrooves were 5 $\mu$m wide, 4 or 5.4 $\mu$m deep, and had a 5 $\mu$m inter-groove spacing. Cells were immunostained for lamin A/C to demarcate the nuclei. Epifluorescence images were acquired with a 20X objective.

**Data Specifications.** 16 bit 2044×2048 pixel single channel images at 0.325 $\mu$m/pixel resolution from six independent experiments were acquired. The dataset contained 137 images of either WT ($n = 73$) or $\Delta$K32 ($n = 64$) myoblast cells. Each image had lamin A/C-stained nuclei of WT or $\Delta$K32 cells as shown in Fig. 1B. The dataset was divided into cross-validation subsets at the experiment level to avoid data leakage due to variations in experimental conditions, such as image focus or illumination.

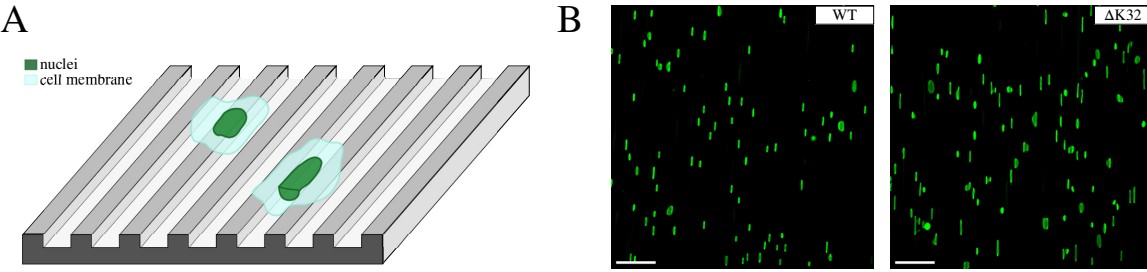

Figure 1: (A) Schematic of cell culture on a microgroove substrate. (B) Sample images of WT and $\Delta$K32 nuclei on microgroove substrates. Scale bar = 100 $\mu$m.

**Preprocessing.** To mitigate the effects of variable cell density, we excluded images with extremely low or high density. To address the limited dataset size, nine 1024×1024 pixel patches were extracted from each image using a sliding window approach with 50% overlap. 99th percentile intensity clippings were applied. Table 1 shows an overview of the dataset.

Table 1: Dataset overview. The average numbers of nuclei per image/patch are balanced.

| Mutation | #images | #patches | Avg #nuclei/image | Avg #nuclei/patch |
|---|---|---|---|---|
| WT \| $\Delta$K32 | 73 \| 64 | 657 \| 576 | 82 \| 87 | 22 \| 24 |

**Experiment setup.** We performed 3-fold cross-validations using patches as inputs, ensuring balanced subsets by pairing experiments based on image quantity. Validation on 15% of the training subset was used to determine the best-performing model based on validation loss. Test results on patches are reported on the test subset. Additionally, we also report test results obtained by labeling directly on whole images, enabled by the adaptive average pooling layer in ResNet-50. This layer ensures consistent fixed-size feature maps, which allows direct evaluation of large images.

We trained ResNet-50 networks (He et al., 2016) from scratch with cross-entropy loss and AdamW optimizer. Patches were downsized to 512×512. Augmentation involved random

flip, zoom, intensity shift, and gamma shift with probabilities of 0.7, 0.3, 0.4, and 0.4, respectively. To mitigate potential bias related to nuclei/groove orientation being associated with label classes, we randomly rotated the patches within the range of [-$\pi$, $\pi$] radians. Test images were padded to 2048×2048 then downsized to 1024×1024. We used test-time augmentations, including horizontal and vertical flipping of the test patches and images.

## 3. Results and Discussion

Table 2 provides the mean Precision, Recall, and F-1 test scores from our 3-fold cross-validation. Our method succeeds in classifying WT and $\Delta$K32 at both patch and image levels. Mean scores are higher with lower variance at the image level due to greater cell density compared to patch evaluations.

Table 2: Average classification test results at patch and image levels for 3-fold cross validation. PR - precision, REC - recall. Avg is the weighted average.

|  | Patch scores | | | Image scores | | |
| --- | --- | --- | --- | --- | --- | --- |
|  | PR | REC | F-1 | PR | REC | F-1 |
| **Avg** | $0.95 \pm 0.04$ | $0.94 \pm 0.05$ | $0.94 \pm 0.05$ | $0.99 \pm 0.02$ | $0.99 \pm 0.02$ | $0.99 \pm 0.02$ |

Fig. 2 depicts activation maps using GradCAM (Selvaraju et al., 2019) on the last convolution layer of our trained ResNet-50. We can observe that the high-activation regions for each class correspond to nuclei with specific deformation patterns. The nuclei activated for the WT class seem to exhibit mild elongation with major axis lengths of up to 20 $\mu$m. In contrast, those activated for $\Delta$K32 exhibit more pronounced elongation, surpassing 30 $\mu$m in length. We also note fewer activated nuclei on the $\Delta$K32 than the WT cases, which suggests the importance of using a large field of view for mutation characterization.

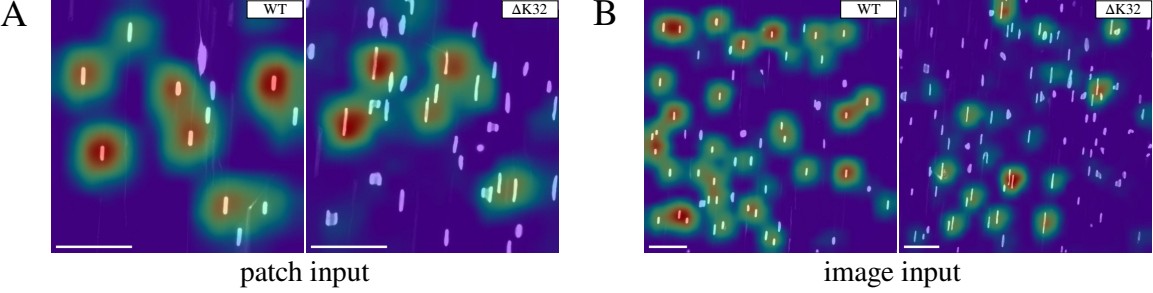

A patch input      B image input

Figure 2: GradCAM visualizations at the patch (A) and image (B) levels of the last convolution layer of our trained ResNet-50 (scale bar = 100 $\mu$m).

## 4. Conclusion

We presented promising classification results for a novel application of ResNet-50 to classify myoblast mutation based on nuclear deformations induced by microgroove substrates. Our models achieved excellent performance in classifying wild type versus mutant myoblasts, with results explainable through GradCAM visualizations. Future work involves applying the method to explore multi-mutation classification and classifying genetic mutations in other cell types such as cancer cells and fibroblasts.

## Acknowledgements

**Funding** This work is funded in part by an endowment in Cardiovascular Bioengineering from the AXA Research Fund (to A.I.B.) and a doctoral fellowship from Institut Polytechnique de Paris (to X.Z.).

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
