# OpenReview forum: "Myoblast Mutation Classification via Microgroove-Induced Nuclear Deformations"
_MIDL.io/2024/Short_Papers — MIDL 2024 Short Papers_

### Official Review · Reviewer_CMfz · 2024-04-24

**Confidence:** 3
**Final Rating:** 5

**Review:**

This paper presents a classification method based on convolutional neural networks to predict myoblast mutation in microscopy images of microgroove-induced nuclear deformations.
The paper is well written and contains a substantial amount of details for such a short paper format.

#### PROS
* Applications of microgroove substrates are relatively recent
* Paper is well written, experimental design is well described
* Results are promising, both at slide and patch level
* Visual feedback from the trained model via GradCAM provides confirms some hypothesis on the learned representation of the ResNet model

#### CONS
* The technical approach is not novel, based on ResNet for image classification. However, the application is novel to the best of my knowledge
* There is no comparison with other approaches

---

### Decision · Program_Chairs · 2024-04-26

Accept